# Anti-Inflammatory Cytokine Profiles in Thrombotic Thrombocytopenic Purpura—Differences Compared to COVID-19

**DOI:** 10.3390/ijms251810007

**Published:** 2024-09-17

**Authors:** Flóra Demeter, György Bihari, Dorina Vadicsku, György Sinkovits, Erika Kajdácsi, Laura Horváth, Marienn Réti, Veronika Müller, Zsolt Iványi, János Gál, László Gopcsa, Péter Reményi, Beáta Szathmáry, Botond Lakatos, János Szlávik, Ilona Bobek, Zita Z. Prohászka, Zsolt Förhécz, Tamás Masszi, István Vályi-Nagy, Zoltán Prohászka, László Cervenak

**Affiliations:** 1Department of Internal Medicine and Haematology, Semmelweis University, 1088 Budapest, Hungary; 2Research Group for Immunology and Hematology, Semmelweis University—HUN-REN-SU (Office for Supported Research Groups), 1085 Budapest, Hungary; 3Department of Haematology and Stem Cell Transplantation, Central Hospital of Southern Pest National Institute of Haematology and Infectious Diseases, 1097 Budapest, Hungary; 4Department of Pulmonology, Semmelweis University, 1083 Budapest, Hungary; 5Department of Anaesthesiology and Intensive Therapy, Semmelweis University, 1085 Budapest, Hungary; 6Department of Infectology, Central Hospital of Southern Pest, National Institute of Haematology and Infectious Diseases, 1097 Budapest, Hungary; 7Department of Anaesthesiology and Intensive Therapy, Central Hospital of Southern Pest, National Institute of Haematology and Infectious Diseases, 1097 Budapest, Hungary

**Keywords:** thromboinflammation, immunothrombosis, thrombotic thrombocytopenic purpura, COVID-19, severe acute respiratory syndrome coronavirus 2 (SARS-CoV-2), inflammation, cytokines

## Abstract

Thromboinflammation/immunothrombosis plays a role in several diseases including thrombotic thrombocytopenic purpura (TTP) and COVID-19. Unlike the extensive research that has been conducted on COVID-19 cytokine storms, the baseline and acute phase cytokine profiles of TTP are poorly characterized. Moreover, we compared the cytokine profiles of TTP and COVID-19 to identify the disease-specific/general characteristics of thromboinflammation/immunothrombosis. Plasma concentrations of 33 soluble mediators (SMs: cytokines, chemokines, soluble receptors, and growth factors) were measured by multiplex bead-based LEGENDplex™ immunoassay from 32 COVID-19 patients (32 non-vaccinated patients in three severity groups), 32 TTP patients (remission/acute phase pairs of 16 patients), and 15 control samples. Mainly, the levels of innate immunity-related SMs changed in both diseases. In TTP, ten SMs decreased in both remission and acute phases compared to the control, one decreased, and two increased only in the acute phase compared to remission, indicating mostly anti-inflammatory changes. In COVID-19, ten pro-inflammatory SMs increased, whereas one decreased with increasing severity compared to the control. In severe COVID-19, sixteen SMs exceeded acute TTP levels, with only one higher in TTP. PCA identified CXCL10, IL-1RA, and VEGF as the main discriminators among their cytokine profiles. The innate immune response is altered in both diseases. The cytokine profile of TTP suggests a distinct pathomechanism from COVID-19 and supports referring to TTP as thromboinflammatory rather than immunothrombotic, emphasizing thrombosis over inflammation as the driving force of the acute phase.

## 1. Introduction

The crosstalk between the innate/adaptive immune system and the hemostatic system has long been recognized [1]. The term ’immunothrombosis’ was initially coined to describe the process by which the immune response contributes to thrombus formation via neutrophil extracellular traps to facilitate the containment and destruction of pathogens [2]. Over time, however, it has become evident that the interplay between inflammation and coagulation is far more complex and extends beyond pathogen-induced conditions to other pathological disorders, including but not limited to venous thromboembolism, tumors, major trauma, and antiphospholipid syndrome [3,4,5]. This understanding has led to the broader application of the term ’immunothrombosis’, which is now commonly used to describe the co-occurrence of microvascular inflammation and thrombosis. The term ‘thromboinflammation’ is often used interchangeably by many for this phenomenon, whereas some use it specifically to refer to the inflammatory processes that develop as a consequence of thrombosis [6].

The concept of immunothrombosis has gained more popularity following the emergence of the COVID-19 pandemic, a pandemic caused by severe acute respiratory syndrome coronavirus 2 (SARS-CoV-2). In COVID-19, a hyperinflammatory state coincides with the overactivation of the coagulation system, highlighting immunothrombosis as the driving force of disease pathology [7]. The clinical presentation of COVID-19 varies widely, ranging from asymptomatic or mild cases to severe and critical ones. Due to excessive inflammation and thrombotic complications, the disease occasionally progresses to acute respiratory distress syndrome (ARDS) and multiple organ failure, with a significant risk of mortality [8]. Prominent features of the COVID-19 pathomechanism, such as complex coagulopathy, diffuse alveolar damage, endothelial dysfunction, and robust immune system activation correlate with disease severity. Hallmarks of immune activation in COVID-19 include complement system activation [9,10,11] and extensive cytokine production, known as the cytokine storm [7,12,13].

Cytokines play a crucial role in the immune system, regulating various biological activities through cell-to-cell communication. They are secreted by a variety of cells and typically act on immune and endothelial cells to trigger diverse biological activities. Cytokine-mediated processes form an extremely complex system, as cytokines exert overlapping and pleiotropic effects, both pro- and anti-inflammatory, on the same biological process, depending on the cellular source, target, and specific phase of the immune response [14,15]. As a given condition is mediated by interactions between different cytokines, a multiplex approach could be more representative than the measurement of a single cytokine [15]. Cytokines are not only mediators of several physiological processes, but also of pathological conditions in which abnormal cytokine production can lead to organ failure or even death. Examining the levels and profiles of these soluble molecules can provide deeper insights into the pathomechanisms of various diseases. In COVID-19, where a hyperinflammatory state is present, the cytokine storm has been extensively studied [16], yet remains highly controversial, with only a few consistent trends emerging from meta-analyses. In contrast, in several other diseases, such as thrombotic thrombocytopenic purpura (TTP), we have little to no information on cytokine profiles.

TTP is a thrombotic microangiopathy characterized by life-threatening acute episodes alternating with phases of remission. It presents with microangiopathic hemolytic anemia and thrombocytopenia, often accompanied by variable neurologic deficits [17]. In immune-mediated TTP, the ADAMTS13 enzyme activity is deficient due to the presence of anti-ADAMTS13 autoantibodies, leading to insufficient breakdown of vWF multimers and consequently to the formation of platelet thrombi without significant coagulation abnormalities. Endothelial cell activation and dysfunction serve as a potential trigger for this process [17]. Several key pathogenic factors and immune mechanisms that drive TTP, including the coexistence of microvascular thrombosis and inflammation, endothelial damage, and excessive complement system activation, are well characterized. In our laboratory, with a long history of TTP research [18,19,20,21], we have extensively studied the role of the complement system in disease pathology [22,23]. Immune-mediated TTP is a B-cell-mediated autoimmune disease, with immunosuppression (most recently first-line rituximab) being the primary treatment approach. However, other aspects of the immune response, such as the potential involvement of various pro- and anti-inflammatory cytokines and other soluble immune mediators, remain poorly understood.

Despite the fact that thromboinflammation/immunothrombosis is often mentioned in the context of both TTP and COVID-19, extensive research has focused primarily on the cytokine storm of COVID-19. In contrast, there are limited data available on the baseline cytokine profiles of TTP patients and only a few studies directly comparing remission to the acute phase; thus, this is our primary focus. Moreover, we aimed to compare the cytokine profiles of TTP and COVID-19 to identify the disease-specific or general characteristics of thromboinflammation/immunothrombosis, thereby potentially enhancing the therapeutic management of these diseases.

## 2. Results

### 2.1. Patient Demographics and Clinical Characteristics

In this retrospective cross-sectional study, we analyzed citrated plasma samples from a cohort of 16 immune-mediated thrombotic thrombocytopenic purpura patients (hereafter referred to as the TTP group) taken both in remission and acute phases, alongside those of 32 hospitalized, non-vaccinated COVID-19 patients (hereafter referred to as the Covid group) categorized into three severity groups (Covid 2, Covid 3, and Covid 4), and 15 age- and sex-matched healthy controls (Figure 1). The Covid 1 group, consisting of outpatients in the convalescent phase [9,10,24], was not included in this study. Median [IQR] age, sex distribution, median [IQR] sampling time (the time elapsed between symptom onset and sampling), and the application of disease-specific therapy in the last 6 months prior to sampling are reported in Table 1 by patient group.

Due to the rarity of TTP and the finite pool of stored COVID-19 samples, we had limited options for sample selection. In addition to this consideration, hospitalized COVID-19 patients tend to be older, and TTP patients tend to be middle-aged. Owing to these factors, a statistically significant difference in age was present between the TTP and Covid groups (Kruskal–Wallis test with Dunn’s post test: *p* = 0.0001). Therefore, the control group was selected to cover the age ranges of both disease groups, resulting in no significant differences in age between the TTP and control (Kruskal–Wallis test with Dunn’s post test: *p* = 0.1643) or the Covid and control (Kruskal–Wallis test with Dunn’s post test: *p* = 0.1942) groups. The age of the Covid 3 group was significantly higher than that of the Covid 2 group (Kruskal–Wallis test with Dunn’s post test: *p* = 0.0005). Nonetheless, no significant correlation was found between age and level of any of the 33 SMs (SMs: cytokines, chemokines, soluble receptors, and growth factors) studied that could explain the differences between the groups (Appendix A). No significant differences were found in sex distribution in the patient groups (Chi^2^ test: *p* = 0.8845).

Covid samples were taken upon the initial admission of patients to either the Central Hospital of Southern Pest—National Institute of Haematology and Infectious Diseases or to Semmelweis University, or upon transfer from another hospital to our centers. Consequently, the time between symptom onset and sampling ranged from 0 to 34 days, with a median of 6 days. The sampling time was significantly longer in the Covid 2 group compared to both the Covid 3 and Covid 4 groups (Kruskal–Wallis test with Dunn’s post test: *p* = 0.0225 and *p* = 0.0228, respectively). Nonetheless, no significant correlation was found between sampling time and level of any of the 33 SMs studied that could explain the differences between the groups (Appendix A). TTP sampling in the acute phase was performed upon admission prior to the initiation of plasma exchange therapy. Of the sixteen acute phase samples, twelve were collected from the patients during their first acute TTP episode, whereas the remaining four were from relapse. Remission samples were consistently obtained during the follow-up period after each acute episode, with a median [IQR] duration of 14.4 [6.5–83] months between the acute and remission samples.

None of the COVID-19 patients included in the study had received any of the following therapies prior to sampling: hydroxychloroquine, tocilizumab, favipiravir, oseltamivir, lopinavir/ritonavir, convalescent plasma, or IVIG. Of the sixteen TTP patients (with the status of one patient unknown), eight had received immunosuppressive therapies (corticosteroids with/without rituximab) at least once within the last 6 months before sampling during the remission phase. The median [IQR] time since the last therapy before sampling was 1.5 [1–2.25] months. At a *p* value threshold of 0.05, levels of sTNF-RII and MCP-1 were significantly higher (Mann–Whitney test: *p* = 0.014 and *p* = 0.0401, respectively) in patients who received therapy within the last 6 months before sampling compared to those who had no therapy during this period (Appendix A). However, after applying a 5% false discovery rate correction using the Benjamini–Hochberg method, these differences were no longer significant. The levels of the remaining 31 SMs showed no significant differences.

### 2.2. Cytokine Profiles of TTP and COVID-19

We measured the concentrations of 33 soluble mediators (SMs: cytokines, chemokines, soluble receptors, and growth factors) in the six patient groups outlined above using a multiplex bead array (Figure 1).

#### 2.2.1. Cytokines under the Detection Limit

Plasma concentrations of six SMs out of the thirty-three—specifically GMCSF, IL-13, IL-1β, IL-2, IL-4, and IL-5—were below the detection limit in at least half of the measured samples (Figure 2 and Appendix A). Consequently, they were excluded from further analysis. CCL4 was only detectable in the Covid groups; therefore, it was analyzed only in the context of COVID-19 and not in TTP.

#### 2.2.2. Cytokine Profile of TTP

In TTP (Figure 2A and Appendix A), the concentration of two SMs, PTX3 and sTNF-RI, were higher in the acute phase compared to remission, and their levels in the latter did not differ from those in the control group. Similarly, among the 11 SMs that decreased in TTP, the concentration of sCD40L was lower only in the acute phase compared to remission, and its level in the latter did not differ from those in the control group. Of the 11 decreased SMs, the remaining 10—primarily pro-inflammatory cytokines (e.g., IFN-γ, IFN-α2, and TNF-α) affecting various cell types, as well as other SMs (e.g., VEGF)—were decreased in both the remission and acute phases compared to the control group. The pattern of seven further cytokines was ambiguous and could be divided into two subgroups. The levels of IL-6, IL-10, CXCL8, GCSF, and CCL3 in the acute phase did not differ from those in the control group, but they were higher than in the remission phase. The other subgroup includes CXCL10 and MCP-1. Their concentrations did not differ from those in the control group in either the remission or acute phase, but they were significantly different from each other, being lower in the acute phase than in the remission phase. To assess the potential impact of immunosuppressive therapy on the seven SMs with ambiguous patterns, we excluded patients who had received any of the aforementioned immunosuppressive therapies within 6 months prior to sampling. This led to the elimination of the previously observed difference in MCP-1 levels between remission and acute phases. However, unlike MCP-1, this exclusion did not resolve the ambiguity of the patterns of the six remaining SMs.

#### 2.2.3. Cytokine Profile of COVID-19

In contrast to TTP, in the Covid group (Figure 2B and Appendix A), eleven SMs were elevated compared to the control group, many of which were important pro-inflammatory factors. However, the concentrations of several other pro-inflammatory cytokines/chemokines showed either no change (e.g., TNF-α, IFN-γ, IFN-α2, IL-7, and IL-15) or a decrease (e.g., CCL3). Of the eleven elevated SMs, ten SMs and the one decreased SM exhibited a stepwise increase or decrease depending on disease severity, a trend hereafter referred to as the severity trend.

#### 2.2.4. Comparing Cytokine Profiles of Acute TTP and Severe COVID-19

To compare the cytokine profiles of acute TTP and COVID-19, the Covid 3 and Covid 4 groups were combined (Covid 3–4), and referred to as severe Covid. The concentrations of 16 SMs, mainly pro-inflammatory mediators, were higher in severe Covid than in acute TTP, whereas only the CCL3 level was higher in acute TTP (Figure 3, and Appendix A).

### 2.3. Principal Component Analysis

To identify the most influential SMs that differentiate two selected patient groups (i.e., control vs. TTP remission, TTP remission vs. acute TTP, control vs. severe Covid, and acute TTP vs. severe Covid), principal component analysis was performed, calculating three principal components (PCs) for each comparison (Figure 4, and Appendix A). Of the three PCs, we selected the one that differentiated between the two studied groups the best, and determined the three SMs that most significantly contributed to this specific PC. In distinguishing the control and TTP remission groups, IL-15, IFN-α2, and IL-12p70 emerged as the three most important SMs based on PC1, with the three PCs collectively explaining 71.5% of the variance. The TTP remission and acute TTP groups were best differentiated by PTX3, CX3CL1, and sTNF-RI, as indicated by PC2, with the three PCs collectively explaining 58.2% of the variance. The primary SMs to distinguish between the control and severe Covid 3–4 groups were sTNF-RI, sCD25, and IL-1RA based on PC1, with the three PCs cumulatively explaining 81.5% of the variance. In contrast, CXCL10, IL1RA, and VEGF were the most discriminating markers for the acute TTP and severe Covid 3–4 groups based on PC1, with the three PCs explaining 67.8% of the cumulative variance.

## 3. Discussion

In this study, we report the distinct soluble mediator profiles of COVID-19 and TTP, two diseases in which immunothrombosis/thromboinflammation plays a pathogenic role. Our findings indicate that, although the alteration of the innate immune response is a common determinant in both diseases, their cytokine profiles exhibit unique characteristics. COVID-19 showed concomitant elevated levels of several pro- and anti-inflammatory soluble mediators, further supporting the established concept of the COVID-19 cytokine storm [7,12,13]. In contrast, the cytokine profile of TTP was characterized by changes in the levels of soluble mediators predominantly pointing towards an anti-inflammatory direction, suggesting a distinct pathomechanism from COVID-19. Our findings support the notion of labeling TTP as thromboinflammatory rather than immunothrombotic, to emphasize thrombosis rather than inflammation as the driving force of the acute phase.

GMCSF, IL-13, IL-1β, IL-2, IL-4, and IL-5 were undetectable in the vast majority of both TTP and COVID-19 samples. Similarly, Shariatmadar et al. observed very low levels of these SMs in patients with acute TTP [25]. Likewise, several previous studies reported very low or undetectable levels of these SMs in COVID-19 patients [26,27,28,29,30,31,32,33,34,35,36]. Apart from IL-1β, which is primarily produced by macrophages [37], these SMs are mainly produced by T lymphocytes and are crucial mediators of the adaptive immune response [38,39]. Their low plasma concentrations do not indicate a massive systemic activation of adaptive immunity in the pathogenesis of either COVID-19 or TTP. Conversely, most of the remaining 27 SMs that were detectable are linked to innate immunity, which appeared to be enhanced in COVID-19 but diminished in TTP. In contrast to COVID-19, where this phenomenon is well known and supported by both cytokine levels and lymphopenia [7,13,40], this is a novel observation in TTP.

The hyperactivation of the immune system and the uncontrolled release of cytokines in COVID-19 became evident early on. Several previous studies have identified key cytokines involved in disease pathology and reported data on the characteristic COVID-19 cytokine storm [12,13,26,30,31,35,41,42,43,44]. On the one hand, we found markedly elevated plasma concentrations of various pro-inflammatory mediators, including IL-1RA, IL-6, IL-18, CCL4, CXCL8, and CXCL10, as well as soluble receptors such as sCD25, sCD40L, sTNF-RI, sTNF-RII, and PTX3, which is consistent with previous findings [26,41,42,43,44,45,46,47,48,49,50]. Soluble receptors contribute to the resolution of inflammation; however, their varying levels can reflect either pro-inflammatory or anti-inflammatory processes, depending on the current phase of the immune response. Apart from sCD40L, all the elevated mediators showed a trend of increasing with disease severity, highlighting their importance in disease pathology. On the other hand, the plasma levels of several other important pro-inflammatory mediators that are commonly associated with inflammatory diseases showed either no change (e.g., TNF-α, IFN-α2, IFN-γ, IL-7, IL-15) or even a decrease (e.g., CCL3). Our finding that TNF-α and IFN-γ are of little importance in COVID-19 is not unprecedented; their concentrations either did not change or showed very small, insignificant increases in several previous studies [31,34,41,42,44]. These slight differences may be due to variations in patient cohorts and measurement methods. These findings highlight the heterogeneous nature of inflammation and its unique manifestations across different diseases, underscoring the need to identify disease-specific soluble mediator profiles for other inflammatory conditions.

Despite the fact that TTP, similarly to COVID-19, is a life-threatening disease involving thromboinflammation, its SM profile is poorly characterized. The course of the disease involves acute episodes alternating with phases of remission, which raises the need to investigate the changes in SM levels across both phases. To our knowledge, this study is the first comprehensive investigation of the soluble mediator profile of TTP, comparing SM levels between remission and acute phases. In contrast to COVID-19, which showed markedly elevated plasma concentrations of 11 SMs, we observed decreased levels of several SMs, mainly pro-inflammatory cytokines (e.g., IFN-γ, IFN-α2, TNF-α, IL-7, IL-17A, and IL-15) in both the remission and acute phases of TTP. Only the concentrations of two soluble receptors (PTX3 and sTNF-RI) were found to be elevated in the acute phase; however, their pro- or anti-inflammatory properties remain unclear, as previously discussed. Interestingly, we previously found that PTX3 elevation was exceptional in TTP, whereas it was elevated in typical and atypical hemolytic uremic syndrome and secondary thrombotic microangiopathies [23]. This contradictory result may be explained by variations in the current phase of the immune response at the time of sampling or other differences between the two patient cohorts. In COVID-19, changes in SM levels concomitantly point to both pro- and anti-inflammatory directions, emphasizing the depletion of the immunoregulatory capacity. This phenomenon is similar to that which is well known in disseminated intravascular coagulation, where hemorrhage and thrombosis occur simultaneously. In contrast to that of COVID-19, the SM profile of TTP suggests a predominance of anti-inflammatory processes. The patterns of seven cytokines (CXCL10, MCP-1, IL-6, IL-10, CXCL8, GCSF, and CCL3) were more complex and difficult to interpret in TTP, which may be due to a variety of reasons, such as the genetic characteristics of the patients or the effect of immunosuppressive therapy. After excluding patients receiving immunosuppressive therapy (corticosteroids with/without rituximab) within 6 months prior to sampling from the analysis, we found that the observed differences in MCP-1 levels between TTP remission and acute TTP groups disappeared. This highlights the fact that immunosuppressive therapy can significantly alter SM concentrations. The ambiguous patterns of the other six SMs can potentially be explained by the small sample size, the effects of therapy, and possibly other factors. Westwood et al. found no significant differences in IL-6 and IL-10 levels between 15 paired acute and remission samples, although elevated IL-6 and IL-10 levels were observed when comparing the 34 acute samples to the 15 remission samples [51].

We observed significantly decreased VEGF levels in both the remission and acute phases of TTP. Moreover, VEGF was one of the three main SMs that differentiated between the severe Covid and acute TTP groups based on PCA. VEGF may have a pathological role, as suggested by previous studies of drug-induced thrombotic microangiopathy (TMA) resulting from treatment with VEGF inhibitors (such as bevacizumab) or VEGF receptor blockers (tyrosine kinase inhibitors, such as pazopanib and sunitinib) [52,53,54,55]. Eremina et al. showed that a local reduction in VEGF in the kidney causes profound thrombotic glomerular injury [56]. Additionally, Mutneja et al. found that high circulating levels of VEGF due to Castleman’s disease suppress glomerular VEGF expression, thereby causing renal TMA [57]. In addition to the general protective role of VEGF on endothelial cell function [58], VEGF was found to play a pivotal role in maintaining the homeostasis of the endothelial–podocyte complex [56,59]. Moreover, it has been hypothesized that diffuse endothelial injury, resulting from the inhibition of VEGF-mediated endothelial protection, contributes to the pathogenesis of TTP [60]. Taken together, VEGF dysregulation appears to be an important pathogenic factor in TTP, underscoring the significance of our finding of decreased VEGF levels in TTP patients.

Principal component analysis can help focus on differences stemming from distinct mechanisms or representing varied effects by identifying the most significant SMs that distinguish between two selected patient groups. Of the four comparisons performed, we could identify a common feature among the three most critical differentiating SMs only in the comparison between the remission and acute phases of TTP. In this case, PTX3, CX3CL1, and sTNF-RI emerged as the most important SMs, all targeting the elimination of inflammation and tissue damage. The decreased level of CX3CL1, an inflammatory mediator that promotes the strong adhesion of leukocytes to activated endothelial cells, suggests the reduced transmigration of inflammatory cells. Moreover, elevated levels of the potent anti-inflammatory mediators PTX3 and sTNF-RI could facilitate debris clearance. In the remaining three comparisons, we were unable to identify a common characteristic to clearly define the three most critical differentiating mediators.

In conclusion, the innate immune response is altered in both COVID-19 and TTP; however, it is altered in differing ways. COVID-19 is marked by concomitantly increased pro- and anti-inflammatory factors, whereas TTP primarily exhibits anti-inflammatory changes. Despite the simultaneous involvement of thrombosis and inflammation in the pathogenesis of TTP, its cytokine profile suggests a distinct pathomechanism from COVID-19. Questioning the practice of using the terms ‘immunothrombosis’ and ‘thromboinflammation’ interchangeably, we propose making a distinction between them. Describing TTP as thromboinflammatory and COVID-19 as immunothrombotic would more accurately capture the primary factors driving their pathogenesis. The simultaneous release of several counteracting SMs in COVID-19 indicates a substantial dysregulation of the immune system, preventing the restoration of the basal state even after viral elimination. In contrast, TTP shows a clear enhancement of the anti-inflammatory immune response. In summary, this suggests that in severe COVID-19, both the inflammatory and coagulation systems need to be restored to their default states. However, in TTP, it may be sufficient to focus on regulating the thrombotic/coagulation system without necessarily interfering with the inflammatory system.

## 4. Materials and Methods

### 4.1. Study Design and Patient Groups

In this retrospective cross-sectional study, we analyzed citrated plasma samples obtained by the centrifugation of peripheral blood from 63 individuals. Samples were collected at the Central Hospital of Southern Pest—National Institute of Haematology and Infectious Diseases or at Semmelweis University, and stored at −80 °C. Measurements were uniformly performed on once-thawed samples for both the cytokine array and enzyme-linked immunosorbent assay (ELISA) investigations.

For the COVID-19 patient groups, SARS-CoV-2 infection was confirmed by qPCR-based testing of mucosal samples, whereas the diagnosis of immune-mediated thrombotic thrombocytopenic purpura was based on the following: (1) one or more episodes of unexplained thrombocytopenia (platelet count below 150 G/L) and microangiopathic hemolytic anemia (Coombs-negative hemolytic anemia, elevated LDH, schistocytes on the blood smear); (2) deficient ADAMTS13 activity (<10%, measured by the FRETS-VWF73 assay); (3) the presence of anti-ADAMTS13 autoantibodies (detected by a functional assay as inhibitors and/or by an anti-ADAMTS13 IgG ELISA). 

The study cohort consisted of 16 immune-mediated TTP patients (hereafter referred to as the TTP group) with samples both from clinical remission and acute phases, and 32 patients with COVID-19 (hereafter referred to as the Covid group) who were hospitalized during the first wave of COVID-19 between April and June 2020, in the pre-vaccine period [10]. Additionally, we included samples from 15 age- and gender-matched healthy controls, recruited from a pool of healthy employees scheduled for mandatory medical check-ups. Hospitalized COVID-19 patients were categorized into 3 severity groups according to their disease status at the time of sampling as follows: Covid 2 (n = 12): no oxygen needed; Covid 3 (n = 13): oxygen support via nasal cannula/mask; and Covid 4 (n = 7): intensive care unit (ICU) treatment. For certain analyses, the latter two groups were combined (Covid 3–4) and referred to as severe Covid. In the Covid 4 group, 6 out of 7 patients and, in the Covid 3 group, 3 out of 13 patients died later. Sample selection aimed for optimal homogeneity in sex and age distribution across all patient groups. Additional criteria included the availability of paired acute and remission samples in the case of TTP patients, whereas for COVID-19 patients, criteria included minimizing the time between symptom onset and sampling and ensuring a balanced number of patients across each severity group.

### 4.2. Cytokine Array

In the cytokine array analysis, we examined 33 soluble mediators (SMs: cytokines, chemokines, soluble receptors, and growth factors) using LEGENDplex™ multiplex bead-based immunoassays. Three different kits, the Human COVID-19 cytokine storm panel 1 (IL-6, MCP-1, GCSF, IFN-α2, IFN-γ, IL-10, CCL3, CXCL10, CXCL8, IL-1RA, IL-7, IL-2, TNF-α), the Human COVID-19 cytokine storm panel 2 (GMCSF, IL-12p70, IL-13, IL-15, IL-17A, IL-18, IL-1β, IL-4, IL-5, CCL4, sCD25, VEGF), and the Human Inflammation panel 2 (CX3CL1, PTX3, sCD25, sCD40L, sRAGE, sST2, sTNF-RI, sTNF-RII, sTREM-1) were utilized according to the manufacturer’s instructions. Briefly, plasma samples were diluted two-fold, whereas the standard series was prepared using a four-fold serial dilution. The reaction was carried out on a filtered 96-well plate. Samples and standards were added to prewetted wells, followed by the addition of a labeled capture bead mix and incubation in the dark at room temperature for 2 h. After vacuum-assisted washing with wash buffer, the detection antibody was added, and the plate was again incubated in the dark at room temperature for 1 h. Streptavidin-PE reagent was then added without washing, followed by a 30 min incubation in the dark at room temperature. After a final vacuum-assisted wash, samples were resuspended in wash buffer. From each tube, 3000 events were recorded using a CytoFLEX (Beckman Coulter, Inc) flow cytometer, and the analysis was performed using the CytExpert software 2.4. We validated our results by repeating the measurement of a selected SM, MCP-1, by sandwich ELISA (see results in Appendix A) and by correlating sCD25 concentrations, measured by two independent Legendplex kits (see results in Appendix A). To validate combining the Covid 3 and Covid 4 severity groups for certain analysis, we correlated their SM concentrations (see results in Appendix A).

### 4.3. Statistical Analysis

Categorical data are reported as counts and percentages. Given that most continuous variables exhibited skewed distributions, data are presented as medians and interquartile (IQ) ranges. Non-parametric statistical tests, including the Mann–Whitney test for two independent groups, the Wilcoxon test for two dependent groups, the Kruskal–Wallis test (with Dunn’s post test) for multiple independent groups, and the Spearman correlation test were carried out using GraphPad Prism v9.5.1 software. *p*-value thresholds were determined after applying a 5% false discovery rate correction using the Benjamini–Hochberg method. Principal component analysis (PCA) was performed on z-scores of concentrations based on soluble mediators that changed significantly between the two groups studied. The number of principal components (PCs) was set to three for each comparison, based on the criterion that the last calculated PC needed to explain at least 10% of the variance in at least one analysis. PCA and the non-parametric Jonckheere test for severity trend were carried out with R v4.4.0 programming language in RStudio v2024.04.1 with the pcaMethods 1.96.0 and PMCMRplus 1.9.10 packages, respectively.

### 4.4. Limitations

The rarity of TTP resulted in a limited pool of paired remission and acute phase samples, restricting the number of TTP patients included in the study. Similarly, the small pool of stored COVID-19 samples further limited the study’s overall sample size. As we analyzed multiple subgroups within both patient groups (based on disease severity for COVID-19 and acute versus remission phases for TTP), the sample sizes for these subgroups were relatively small. Additionally, due to restricted sample selection, certain patient parameters, such as comorbidities and hematological or complement data, were not consistently available across the cohort.

## Figures and Tables

**Figure 1 ijms-25-10007-f001:**
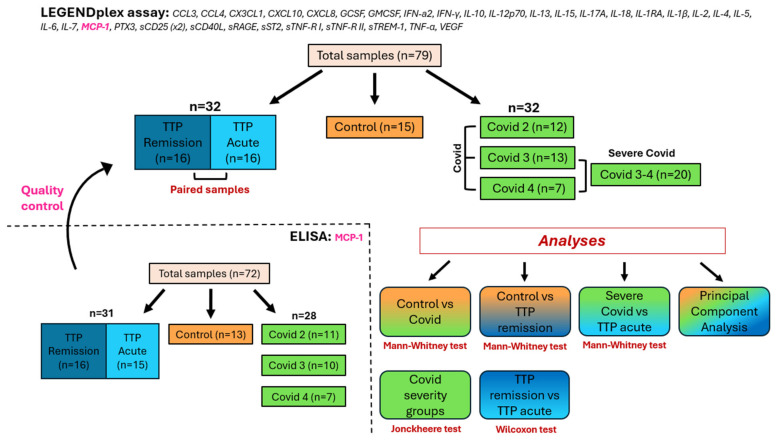
Study design, patient cohorts, and analyses performed. In this study, we analyzed 33 soluble mediators using the LEGENDplex™ assay and validated our results by repeating MCP-1 measurements with sandwich ELISA. The cohort included 16 immune-mediated TTP patients with samples from both remission and acute phases, 32 COVID-19 patients, and 15 age- and gender-matched healthy controls. COVID-19 patients were categorized into three severity groups as follows: Covid 2 (n = 12) with no oxygen need, Covid 3 (n = 13) with oxygen support, and Covid 4 (n = 7) requiring ICU treatment. For certain analyses, the latter two groups were combined (Covid 3–4) and referred to as severe Covid. Patient groups are color-coded throughout the manuscript as follows: the control group in orange, TTP groups in blue, and COVID-19 groups in green, across all figures and tables. The Mann–Whitney, Wilcoxon, and Jonckheere tests, along with principal component analysis, were used to compare the levels of soluble mediators among patient groups.

**Figure 2 ijms-25-10007-f002:**
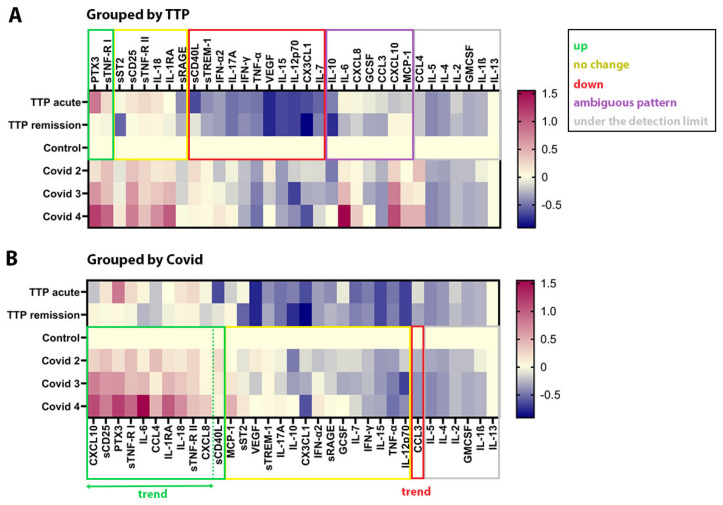
**Soluble mediator profiles of TTP and COVID-19.** The same ratios of the logarithmically transformed median concentration values of the 33 SMs relative to control were plotted twice, each time using a different grouping method for better understanding. In (**A**), the SMs are grouped based on their changes in TTP (for further details, see Section 2.2 of the Results section), whereas in (**B**), they are grouped according to their changes in COVID-19 (ordered according to decreasing value of Covid median/control median in each group), as determined by statistical tests (see Appendix A). Green boxes indicate elevated SM levels compared to control, red boxes signify decreased levels, yellow boxes represent no change, gray boxes show SMs below the detection limit in TTP (**A**) or COVID-19 (**B**), and the purple box indicates an ambiguous pattern.

**Figure 3 ijms-25-10007-f003:**
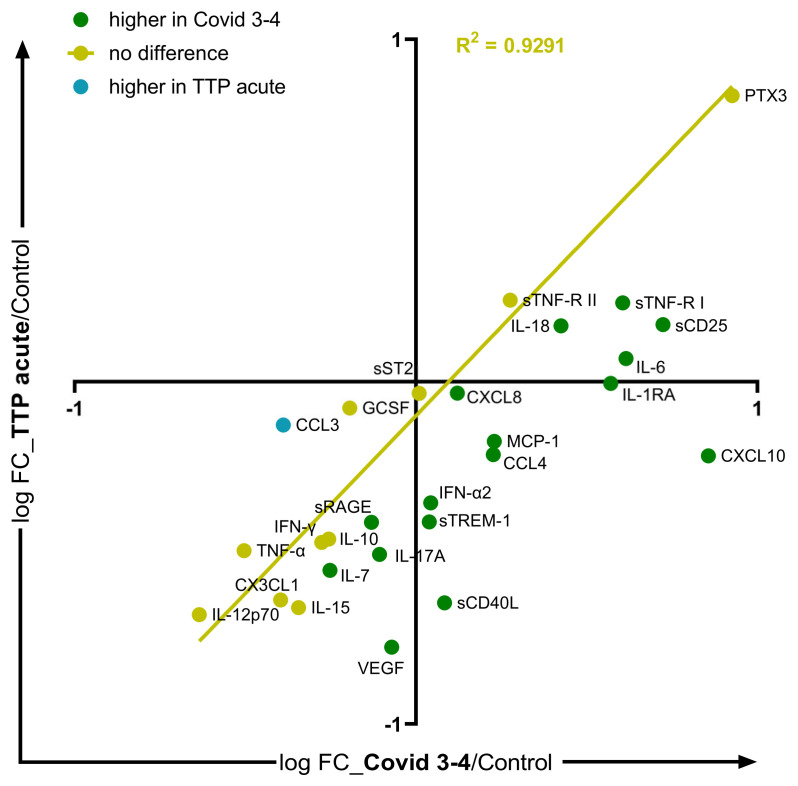
**Comparison of soluble mediator profiles of acute TTP and severe COVID-19.** For the comparison of the cytokine profiles of acute TTP and COVID-19, the Covid 3 and Covid 4 groups were combined (Covid 3–4), and referred to as severe Covid. The logarithmically transformed fold change (FC) values (acute TTP-to-control ratio and Covid 3–4-to-control ratio) of the 33 SMs were plotted. SMs higher in Covid 3–4 than in acute TTP are plotted in green, those higher in acute TTP are in blue, and those with no significant difference are in yellow, based on Mann–Whitney tests. Simple linear regression for the SMs with no significant difference between the two disease groups is plotted.

**Figure 4 ijms-25-10007-f004:**
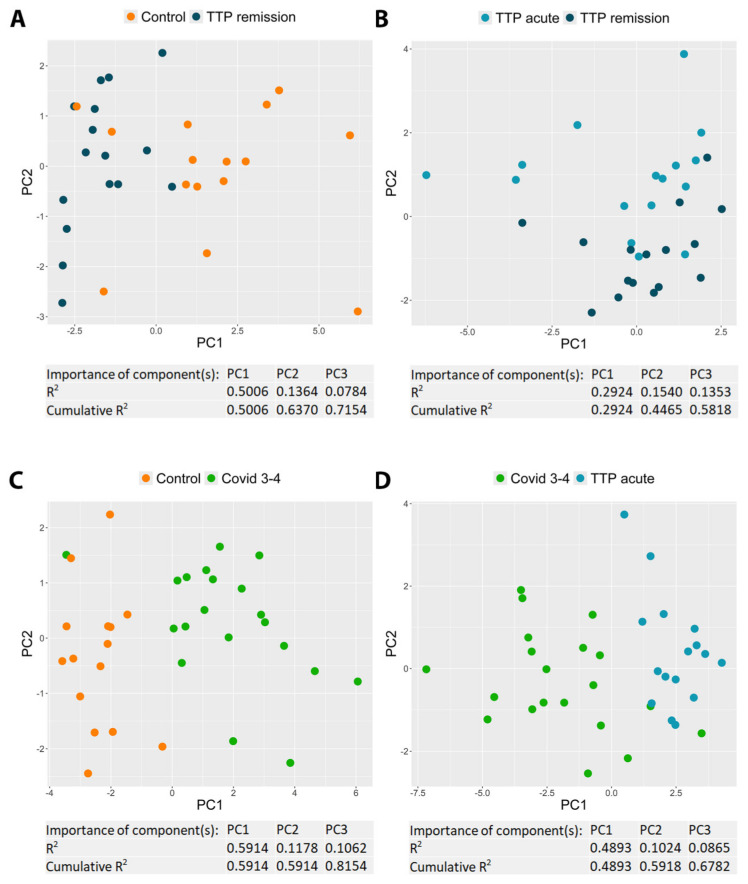
**Principal component analysis.** Principal component analysis was performed on z-scores of concentrations, based on soluble mediators (SMs) which were significantly altered between the control and TTP remission (**A**), TTP acute and TTP remission (**B**), control and Covid 3–4 (**C**), and Covid 3–4 and TTP acute (**D**) groups. Three principal components (PCs) were calculated (and their R^2^ values were plotted), of which the first two were visualized. The importance values of the SMs in each PC are shown in Appendix A.

**Table 1 ijms-25-10007-t001:** Demographics, time between symptom onset and sampling (sampling time), and immunosuppressive therapy within the last 6 months before sampling of TTP and COVID-19 patients and controls. na = not applicable.

	Control	TTP	COVID-19			
			Covid 2	Covid 3	Covid 4	∑ Covid
**Age in years, median [IQR]**	55 [52–65]	43 [34–65]	54 [40.5–76]	82 [70–87]	72 [48–77]	69 [54–87]
**Patient, n**	15	16	12	13	7	32
**Male, n (%)**	8 (53)	6 (37.5)	7 (58)	7 (54)	3 (43)	17 (53)
**Female, n (%)**	7 (47)	10 (62.5)	5 (42)	6 (46)	4 (57)	15 (47)
**Sampling time in days, median [IQR]**	na	na	11 [8–34]	5 [2–34]	3 [2.5–30]	6 [3–34]
**Immunosuppressive therapy within the last 6 months before sampling, n (%)**	na	8/15 (53.3)	0 (0)	0 (0)	0 (0)	0 (0)

## Data Availability

Datasets generated during and/or analyzed in the current study are available from the corresponding author upon reasonable request.

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
