# Peer review of "Anti-Inflammatory Cytokine Profiles in Thrombotic Thrombocytopenic Purpura—Differences Compared to COVID-19"

_ijms, 2024, doi:10.3390/ijms251810007_

Round 1
Reviewer 1 Report
Comments and Suggestions for Authors
Dear Editor
Many thanks for considering me as a potential reviewer for the article "Anti-inflammatory cytokine profile in thrombotic thrombocytopenic purpura differences compared to COVID-19". No doubt the article is well-structured, well-presented and written. However, I have several observations, which I believe should be taken into consideration before proceeding further.
My observations are as follows.
My observations
· If the authors agree, please add ‘s’ to the cytokine in the title, because it sounds better when is ‘cytokine profiles’ and also they use ‘cytokine profiles’ in abstract line 23. My suggestion is; ‘Anti-inflammatory cytokine profiles in thrombotic thrombocytopenic purpura – differences compared to COVID-19’,
· Please cite every piece of information throughout the introduction especially, lines 43-55’,
· Figure 1, needs to be described well,
· Shariatmadar (line-268) should not be italic,
· Where is the citation? (Line 280-281) Several previous studies identified key cytokines involved in disease pathology and reported data on the characteristic COVID-19 cytokine storm.,
· Why Table 1 is highlighted with different colors?, please have a look.
Comments on the Quality of English LanguageDear Editor/authors,
No, doubt the article is well-presented and written.
However, my opinion regarding the English language is; that this manuscript needs minor editing several minor issues were observed.
Thanks
Reviewer 2 Report
Comments and Suggestions for Authors
The article refers to the effect of cytokine in thrombocytopenic purpura in patients infected with SARS CoV2. The rationale is adequate, and the methodology is valid. Some elements need to be clarified. There is limited information regarding patients, age, gender, comorbidities, and treatment. The span of the time when the samples were taken is critical, and the longitudinal analysis is based on previous conditions. Are there alterations in the complement pathway? However, taken as a whole, not adjusted information based on the above variables, the results are important for clinical studies and can be considered a pilot analysis. Pentraxin 3 and CCL3 are the clear primary markers to follow; however, hematological levels of the patients should be included in the primary component analysis as CCL3, see DOI: 10.1371/journal.ppat.1000755. The authors should better discuss Figure 4, The article lacks conclusions and limitations and should be separate items.
Comments on the Quality of English LanguageSeveral grammatical mistakes were encountered
